# Resveratrol Activates Natural Killer Cells through Akt- and mTORC2-Mediated c-Myb Upregulation

**DOI:** 10.3390/ijms21249575

**Published:** 2020-12-16

**Authors:** Yoo-Jin Lee, Jongsun Kim

**Affiliations:** 1Department of Microbiology and Institute for Immunology and Immunological Diseases, Yonsei University College of Medicine, Seoul 08758, Korea; yjlee0610@yuhs.ac; 2Brain Korea 21 PLUS Project for Medical Science, Yonsei University College of Medicine, Seoul 08758, Korea

**Keywords:** NK cells, resveratrol, Akt, mTOR complex 2, c-Myb

## Abstract

Natural killer (NK) cells are suitable targets for cancer immunotherapy owing to their potent cytotoxic activity. To maximize the therapeutic efficacy of cancer immunotherapy, adjuvants need to be identified. Resveratrol is a well-studied polyphenol with various potential health benefits, including antitumor effects. We previously found that resveratrol is an NK cell booster, suggesting that it can serve as an adjuvant for cancer immunotherapy. However, the molecular mechanism underlying the activation of NK cells by resveratrol remains unclear. The present study aimed to determine this mechanism. To this end, we investigated relevant pathways in NK cells using Western blot, real-time polymerase chain reaction, pathway inhibitor, protein/DNA array, and cytotoxicity analyses. We confirmed the synergistic effects of resveratrol and interleukin (IL)-2 on enhancing the cytolytic activity of NK cells. Resveratrol activated Akt by regulating Mammalian Target of Rapamycin (mTOR) Complex 2 (mTORC2) via phosphatase and tensin homolog (PTEN) and ribosomal protein S6 kinase beta-1 (S6K1). Moreover, resveratrol-mediated NK cell activation was more dependent on the mTOR pathway than the Akt pathway. Importantly, resveratrol increased the expression of c-Myb, a downstream transcription factor of Akt and mTORC2. Moreover, c-Myb was essential for resveratrol-induced NK cell activation in combination with IL-2. Our results demonstrate that resveratrol activates NK cells through Akt- and mTORC2-mediated c-Myb upregulation.

## 1. Introduction

Natural killer (NK) cells are innate immune lymphocytes that play a key role in the first line of defense against pathogens and cancer [1]. As the name suggests, NK cells spontaneously kill target cells, such as cancer and virus-infected cells, without presensitization [2]. In addition, NK cells have the potential to regulate T cell responses by secreting cytokines and activating dendritic cells [3]. Naïve NK cells can be activated by various proinflammatory cytokines, such as interleukin (IL)-2, IL-12, and IL-15 [4,5]. These cytokines stimulate NK cell proliferation and enhance NK cell cytotoxicity and have been used in clinical studies to activate NK cells, such as in IL-2 therapy [6]. However, a high-dose IL-2 therapy can lead to severe adverse effects, including vascular leakage and organ injury caused by the activation of the vascular endothelium [6,7].

Cancer immunotherapy is a type of therapy that boosts the natural immune system of the body to fight cancer [8,9,10,11]. Most immunotherapeutic strategies have mainly focused on T cell responses, but recent studies have attempted to use NK cells as potential therapeutic targets in cancer therapy [12,13,14,15]. Despite major breakthroughs, clinical studies have exposed the limitations associated with using NK cells, such as their low treatment efficacy and response rate and resistance to immunotherapies [16,17,18]. To overcome these limitations, researchers are actively searching for new drug candidates that can boost the therapeutic efficacy of cancer immunotherapy [19,20,21,22]. Food nutrients, such as resveratrol, curcumin, and pomegranate, are intriguing sources of therapeutic adjuvants for cancer prevention and treatment because they have relatively favorable safety profiles and have shown promising efficacy for combating tumor cells [23,24].

Resveratrol is a well-studied polyphenol, which has been shown to possess various beneficial properties, including antitumor, anti-inflammatory, antiviral, antifungal, and antibacterial effects [25]. The antitumor effects of resveratrol have been demonstrated in several studies [26]. Interestingly, several studies have reported the synergistic antitumor effect of resveratrol and doxorubicin on different breast and colon cancer cell lines [27,28]. Prior studies have also demonstrated that resveratrol exerts simultaneous effects on NK and T cells [29]. In a previous study, we showed that resveratrol contributes to enhanced NK cell cytotoxicity in vitro and in vivo, suggesting that resveratrol could be used as an adjuvant for cancer immunotherapy [30]. However, the molecular mechanism through which resveratrol activates NK cells remains unclear. Therefore, in this study, we aimed to elucidate the mechanism of NK cell activation by resveratrol.

## 2. Results

### 2.1. Synergistic Effect of Resveratrol and IL-2 on NK Cell Activation

IL-2, IL-12, and IL-15 are known to induce interferon (IFN)-γ secretion and play an important role in the activation of NK cells [4,5]. In a previous study, we showed that resveratrol activates NK cells in a synergistic manner with IL-2 [30]. To identify whether resveratrol has synergistic effects with other cytokines that mediate NK cell activation, we first measured IFN-γ secretion and NK cytotoxicity following treatment with resveratrol and IL-2, IL-12, or IL-15. Among these cytokines, IL-2 produced the highest level of IFN-γ secretion at the same concentration compared to other cytokines when they were added to cells in combination with resveratrol (Figure 1a). In addition, we found that resveratrol treatment in combination with 5 ng/mL IL-2 was as effective as treatment with 10 ng/mL of IL-2 alone (Figure 1a). We next investigated whether resveratrol has synergistic effects with the activating cytokines on NK cell cytotoxicity. As shown in Figure 1b, only the combination of resveratrol and IL-2 showed synergistic effects on the enhancement of the cytolytic activity of NK cells. These results suggest that resveratrol may activate a factor downstream of the IL-2 signaling pathway.

### 2.2. Regulation of the Akt and Mammalian Target of Rapamycin (mTOR) Complex 2 (mTORC2) Signaling Pathway by Resveratrol

IL-2 signaling is propagated following receptor–ligand engagement, controlling the recruitment of Janus kinases 3 (JAK3) and activation of Akt, extracellular signal-regulated kinase (ERK) 1/2, and transcriptionally active signal transducer and activator of transcription 5 (STAT5) [31,32]. Thus, we investigated which molecule(s) is regulated by resveratrol among these IL-2 signaling-mediated proteins. As shown in Figure 2, we found that resveratrol increased the phosphorylation of Akt, but it had no effect on the phosphorylation of Stat5 and ERK. These results suggest that resveratrol activates the Akt signaling pathway but not the Stat5 and ERK signaling pathways.

Next, to determine the mechanism through which resveratrol activates Akt, we measured the expression levels of phosphoinositide-dependent kinase-1 (PDK1) and mTORC2, which are upstream of Akt [33]. As shown in Figure 3a, PDK1 expression was not affected by resveratrol, whereas the level of phospho-rictor (a subunit of mTORC2) decreased significantly. Previous studies have shown that phosphorylated rictor negatively regulates mTORC2 expression as part of a negative feedback mechanism controlling Akt activity [34]. Consistent with this, resveratrol also appeared to inhibit phosphatase and tensin homolog (PTEN) and ribosomal protein S6 kinase beta-1 (S6K1) activities, which are known to downregulate mTORC2 signaling (Figure 3b). Collectively, our data suggest that resveratrol activates Akt by regulating rictor phosphorylation and mTORC2 via PTEN and S6K1.

### 2.3. Effects of Akt and mTORC2 on Resveratrol-Induced NK Cell Activation

We investigated whether NK cell activation is the direct effect of resveratrol-induced Akt and mTORC2 activation. For this purpose, IFN-γ secretion and NK cell cytotoxicity were measured after treatment with the Akt inhibitor MK-2206 (Figure 4a). As shown in Figure 4a, the Akt inhibitor decreased IFN-γ secretion and NK cell cytotoxicity, but the inhibition was to a substantial extent overcome by resveratrol treatment. These results suggest that resveratrol may also activate NK cells in a manner other than the Akt signaling pathway. We next compared IFN-γ secretion and NK cell cytotoxicity after treatment with the mTOR inhibitor KU-0063794 (Figure 4b). As shown in Figure 4b, the mTOR inhibitor decreased IFN-γ secretion and NK cell cytotoxicity, but the inhibition was only slightly overcome by resveratrol treatment. Interestingly, the mTOR inhibitor decreased resveratrol-induced NK cell activation more than the Akt inhibitor (Figure 4a,b). These results suggest that resveratrol-mediated NK cell activation is more dependent on the mTOR pathway than the Akt pathway.

### 2.4. Upregulation of Akt-Related Transcription Factors in NK Cells by Resveratrol

Next, we performed a high-throughput analysis of transcription factor activation using a protein/DNA array to identify the transcription factors activated by resveratrol. As shown in Figure 5a, the activities of 56 transcription factors in NK cells were screened using this method. To select candidates whose activity increased following resveratrol treatment, we first excluded candidates with a spot density lower than 4000. Next, we selected transcription factors with a higher than 2-fold increase in spot density induced by resveratrol compared with treatment with IL-2 alone (Figure 5a). Consequently, we identified four transcription factors (CBF, c-Myb, NF-E2, and SP-1) and found that they were all related to the Akt signaling pathway [35,36,37,38]. Resveratrol is known to upregulate NF-E2 expression, and we previously reported that c-Myb is a key factor involved in the regulation of NK cell activity [39,40]. In this study, we further investigated the effect of resveratrol on the expression of c-Myb. As shown in Figure 5b,c, IL-2 appeared to increase the mRNA and protein expression levels of c-Myb. As expected, resveratrol significantly increased the level of c-Myb expression compared with that induced by IL-2 treatment alone. Next, we investigated whether resveratrol-induced c-Myb activation was dependent on the Akt and mTOR pathways. As shown in Figure 5d, the Akt inhibitor decreased c-Myb expression, but the inhibition was overcome by resveratrol treatment. Furthermore, the mTOR inhibitor decreased c-Myb expression, but the inhibition was not completely overcome by resveratrol treatment. Altogether, these results suggest that the mTOR pathway is more important in resveratrol-induced c-Myb upregulation than the Akt pathway.

### 2.5. Effects of c-Myb on Resveratrol-Induced NK Cell Activation

Finally, we comprehensively investigated the extent to which resveratrol is dependent on c-Myb to activate NK cells. To identify whether the c-Myb binding site exists within the IFN-γ promoter region, we used the evolutionary conservation of genomes (ECR) browser, a tool for visualizing and accessing data from comparisons of multiple vertebrate genomes. As shown in Figure 6a, it was predicted that the c-Myb binding site exists within the IFN-γ promoter region. Next, we measured the levels of IFN-γ (Figure 6b) and cytolytic activity (Figure 6c) after treatment with the c-Myb inhibitor, celastrol, to investigate whether resveratrol depends on c-Myb to activate NK cells. As shown in Figure 6b,c, the effects of resveratrol were completely abrogated upon celastrol-induced c-Myb inhibition. We further confirmed that resveratrol action depends on c-Myb by performing c-Myb knockdown studies in NK cells. As shown in Figure 6d, the effects of resveratrol were completely abrogated by using a c-Myb shRNA in the NK92 cell line. Altogether, these results indicate that resveratrol-induced activation of NK cells is dependent on c-Myb.

## 3. Discussion

NK cells are endowed with potent cytolytic activity against tumors, and the efficacy of NK cell-mediated immunotherapy can be enhanced by immune stimulants, such as cytokines and antibodies [4]. In this study, we found that resveratrol showed a synergistic effect in combination with IL-2, IL-12, and IL-15 on IFN-γ secretion; however, a synergistic effect on the cytolytic activity of NK cells was only observed in combination with IL-2 (Figure 1). Additionally, we demonstrated that resveratrol activates Akt by regulating mTORC2 via PTEN and S6K1 (Figure 2 and Figure 3). Moreover, we found that resveratrol-mediated NK cell activation was more dependent on the mTOR pathway than the Akt pathway, and resveratrol upregulated the expression of c-Myb, a key regulator of NK cell activity (Figure 4, Figure 5 and Figure 6). mTORC2 directly activates PKC and SGK; SGK gene family members are known to be similar to those of the MYB gene family [41]. Interestingly, SGK1 is known to regulate Th1 and Th2 differentiation downstream of mTORC2 [42]. Therefore, it would be interesting to confirm whether mTORC2 can also directly activate c-Myb, as in the case of SGK1.

MicroRNAs (miRNAs) are a highly conserved class of small noncoding RNAs with important regulatory functions in cell proliferation, differentiation, signal transduction, immune response, and carcinogenesis [43,44]. miR-155 enhances IFN-γ production in NK cells, and it is upregulated by the transcription factor c-Myb [45,46]. miR-150 is widely expressed in immune cells and plays an important role in the development of lymphocytes and hematopoietic malignancies [47]. Several studies have reported that miR-150 negatively regulates CD8 T cell memory formation and controls B cell differentiation by targeting c-Myb [48,49]. In this study, we found that resveratrol activates NK cells by upregulating the expression of c-Myb (Figure 6). Thus, it would be informative to investigate whether resveratrol activates NK cells by stimulating miR-155 function as a result of upregulating c-Myb expression or by inhibiting miR-150 function.

Tumor progression usually leads to NK cell exhaustion, thus limiting the antitumor potential of NK cells [50]. Tim-3 is involved in T cell exhaustion. Interestingly, Tim-3 is also expressed in NK cells [51]. Tim-3 expression in NK cells increases as the melanoma stage progresses [51]. Furthermore, Tim-3 blockade reverses the exhausted phenotype of NK cells [51]. These data may lead to the development of new therapies targeting Tim-3 in tumor immunotherapy. We previously reported that IFN-γ secretion decreased in NK cells collected from tumor-bearing mice. This phenomenon could have occurred because those NK cells were already activated in vivo, and at the time of sample collection, they were in an exhaustion phase. Furthermore, we observed that resveratrol increases IFN-γ secretion even in NK cells from tumor-bearing mice, suggesting that resveratrol reverses NK cell exhaustion and overcomes the exhaustion-related decreased antitumor potential of NK cells. Thus, it would be worthy to investigate if resveratrol blocks Tim-3 or other exhaustion markers in NK cells from tumor-bearing mice.

Our findings demonstrated that resveratrol enhances NK cell activity by increasing the expression of c-Myb (Figure 6). c-Myb is already known to enhance the function of CD8^+^ T and NK cells. Hence, resveratrol could serve as a potential adjuvant for cancer immunotherapy, since it enhances the CD8^+^ T cell response and NK cell activity.

## 4. Materials and Methods

### 4.1. Cell Lines and Cell Culture

All cell lines were obtained from ATCC (Manassas, VA, USA). The human NK cell line, NK92, was maintained in minimum essential medium α (Gibco, New York, NY, USA) supplemented with 12.5% heat-inactivated fetal bovine serum (FBS; Gibco), 12.5% heat-inactivated horse serum (Gibco), 0.2 mM myo-inositol (Sigma-Aldrich, St. Louis, MO, USA), 0.1 mM 2-mercaptoethanol (Sigma-Aldrich), 0.02 mM folic acid (Sigma-Aldrich), 1% penicillin/streptomycin (WelGENE, Gyeongsansi, Korea), and 5 ng/mL IL-2 (ATGen, Sungnamsi, Korea). The human erythroleukemia cell line, K562, was maintained in RPMI 1640 medium (Gibco) supplemented with 10% FBS and 1% penicillin/streptomycin.

### 4.2. Antibodies and Reagents

Resveratrol, anti-β-actin antibody, and the c-Myb inhibitor, celastrol, were purchased from Sigma-Aldrich. For Western blotting, monoclonal antibodies against phospho-STAT5, STAT5, phospho-Akt, Akt, phospho-Erk, Erk, phospho-PDK1, PDK1, phospho-rictor, rictor, and c-Myb were purchased from Cell Signaling Technology (Danvers, MA, USA).

### 4.3. Enzyme-Linked Immunosorbent Assay (ELISA)

An IFN-γ ELISA set was purchased from BD Biosciences (San Diego, CA, USA). NK92 cells (5 × 10^4^ cells/well) were dispensed in a 24-well plate and incubated for 36 h. Then, the supernatants were harvested and dispensed in triplicate into a 96-well microplate pre-coated with the capture antibody. After incubation for 2 h, each well was washed with washing buffer (0.05% Tween 20 in PBS, pH 7.4), followed by incubation with the HRP-conjugated detection antibody for 1 h. The substrate solution 3,3′,5,5′-tetramethylbenzidine was added to the wells in the dark. Absorbance was measured at 450 nm using a microplate reader (Epoch, BioTek, Winooski, VT, USA).

### 4.4. Cytotoxicity Assay

The calcein-AM (Invitrogen, Carlsbad, CA, USA) assay was used to measure NK cell cytotoxicity against target cells. Calcein-AM was added to target cells at a final concentration of 2 μM and incubated for 30 min at 37 °C and 5% CO_2_. The calcein-AM-labeled target cells were washed with PBS twice, following which 1 × 10^4^ cells were dispensed in quadruplicate into a 96-well round-bottom plate. NK cells were added at different effector:target ratios and co-cultured for 4 h. The amount of calcein-AM released from the lysed target cells was measured using a spectrophotometer (ex: 485 nm/em: 535 nm). Total specific lysis was calculated using the following formula:$$\mathrm{Specific}\;\mathrm{cytotoxicity}\;\left(\%\right)=\;\frac{\mathrm{Experimental}\;\mathrm{release}-\mathrm{spontaneous}\;\mathrm{release}}{\mathrm{Maximum}\;\mathrm{release}-\mathrm{spontaneous}\;\mathrm{release}}\;\times \;100$$

### 4.5. Western Blot Analysis

After treatment, NK92 cells were harvested and washed. The cells were lysed in RIPA buffer supplemented with a protease and phosphatase inhibitor cocktail. Protein concentration was determined using the bicinchoninic acid (BCA) protein assay (Thermo Scientific, Seoul, Korea). Sample buffer was added to each sample, following which the samples were boiled at 100 °C for 5 min, cooled on ice, and loaded onto a 10% sodium dodecyl sulfate (SDS)-polyacrylamide gel. After electrophoresis, the proteins were transferred onto a methanol-activated polyvinylidene fluoride (PVDF) membrane, which was blocked with 5% skim milk to prevent nonspecific reactivity. The membrane was probed with specific antibodies diluted in Tris Buffered Saline (TBS)-T containing 3% bovine serum albumin (BSA) and incubated overnight at 4 °C. The membrane was washed three times for 10 min each and incubated for 1 h with the secondary antibodies. Immunoreactivity was detected using an ECL solution (Thermo Scientific, Seoul, Korea).

### 4.6. Protein/DNA Array for Transcription Factors

The nuclear extracts from NK92 cells were prepared using the nuclear extraction kit (Panomics, Fremont, CA, USA) following the manufacturer’s instructions. A total of 5 μg of nuclear extracts were mixed with the DNA probe (10 ng/μL). Each array was performed following the procedure in the Protein/DNA Array kit (Panomics) user manual. The membrane was detected using chemiluminescent reagents and exposed to photographic film. The array images were acquired, and the spot intensities of all the detected transcription factors were measured using Image J software (java version, NIH).

### 4.7. Real-Time Polymerase Chain Reaction (PCR) Analysis

Total RNA was extracted from NK92 cells using TRIzol (Life Technologies, Carlsbad, CA, USA), and cDNA was synthesized using the SensiFAST cDNA synthesis kit (Bioline, Taunton, MA, USA) according to the manufacturer’s instructions. The synthesized cDNAs were used as templates for subsequent PCR amplification of the c-Myb and GAPDH genes. The following primers were used:c-Myb, 5′-CATGTTCCATACCCTGTAGCG-3′ and 5′-TTCTCGGTTGACATTAGGAGC-3′;GAPDH, 5′-CAGCCTCAAGATCATCAGCA-3′ and 5′-GTCTTCTGGGTGGCAGTGAT-3′.

Real-time PCR was carried out using the KAPA SYBR FAST qPCR kit (KAPA Biosystems, Wilmington, MA, USA), and amplification was performed on an ABI Prism StepOnePlus^TM^ detection system (Applied Biosystems, Foster City, CA, USA) according to the conditions recommended by the manufacturer. The experiments were performed in triplicate, and results were normalized to the expression of GAPDH. The relative expression levels of the target genes were calculated using the 2^−ΔΔCt^ method.

### 4.8. ECR Browser

To identify whether the c-Myb binding site exists within the IFN-γ promoter region, we used the ECR browser, a tool for visualizing and accessing data from comparisons of multiple vertebrate genomes. Annotations of the conserved transcription factor binding sites underlying the ECR Browser conservation plots were displayed using the “Synteny/Alignments” link in the top menu and performed following the procedure provided in the instructions.

### 4.9. c-Myb Knockdown Using Lentiviral Transduction

The MYB shRNA and control shRNA plasmids were purchased from Sigma-Aldrich. Lentiviral particles were produced using a third-generation packaging system. For this system, pMDLg/pRRE (plasmid #12251; Addgene, Cambridge, MA, USA), pRSV-Rev (plasmid #12253; Addgene), and pMD2.G (plasmid #12259; Addgene) were kindly provided by Dr. Didier Trono. To transfect the plasmids into HEK293T cells, Lipofectamine 3000 was used (Invitrogen). For lentiviral transduction, NK-92 cells were stimulated with IL-2 (5 ng/mL) for 1 h and then infected by mixing with the supernatants containing the lentiviral particles and protamine sulfate (15 μg/mL; Sigma Aldrich, Seoul, Korea). To increase lentiviral transduction efficiency, the mixtures were centrifuged at 360× *g* for 90 min at 32 °C. For the selection of transgene-positive cells, the cells were cultured in complete growth medium containing 2 μg/mL puromycin for up to 4 weeks.

### 4.10. Statistical Analysis

Statistical analysis was performed using the GraphPad Prism software (GraphPad software, San Diego, CA, USA). Data are expressed as the mean ± standard error of the mean (SEM). The student’s *t*-test and one- or two-way ANOVA were used to compare distributions between groups. *p* < 0.05 was considered statistically significant.

## Figures and Tables

**Figure 1 ijms-21-09575-f001:**
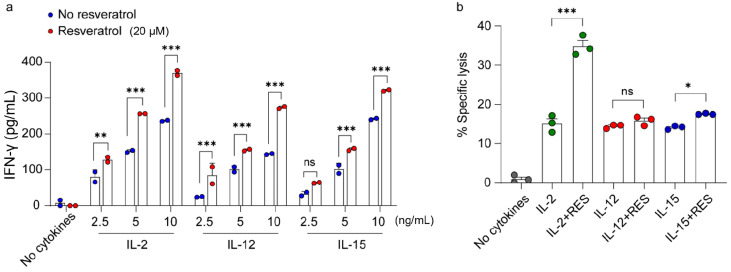
Synergistic effect of resveratrol and interleukin (IL)-2 on natural killer (NK) cell activation. Fresh medium was used without IL-2 for the experiments with IL-12 or IL-15. (**a**) Interferon (IFN)-γ secretion in NK92 cells after the indicated treatments at various concentrations for 48 h determined using an enzyme-linked immunosorbent assay. Data are shown as the mean ± SEM of two independent experiments. (**b**) NK cell cytotoxicity was determined using the calcein-AM assay. NK92 effector cells were treated with each cytokine (IL-2, 5 ng/mL; IL-12 and IL-15, 10 ng/mL) with or without 20 μM resveratrol for 36 h, followed by incubation with K562 target cells at a 1:1 ratio for 4 h. Data are shown as the mean ± SEM of three independent experiments. Asterisks indicate statistical significance using one-way ANOVA: * *p* < 0.05, ** *p* < 0.01, *** *p* < 0.001, ns: not significant (*p* > 0.05).

**Figure 2 ijms-21-09575-f002:**
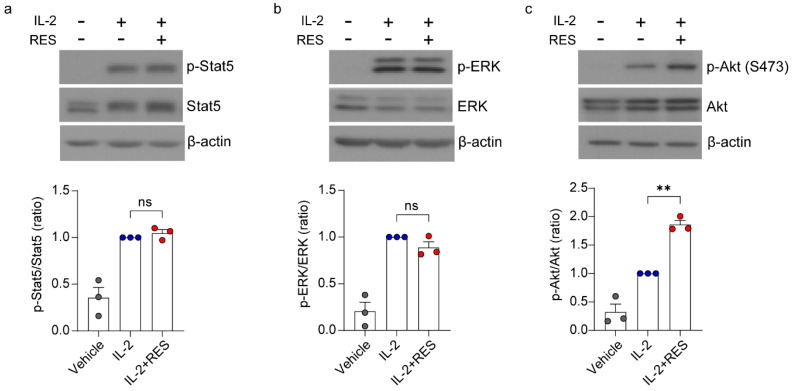
Regulation of the Akt signaling pathway by resveratrol. NK92 cells were deprived of interleukin (IL)-2 for 24 h and then treated with IL-2 (5 ng/mL) with or without 20 μM resveratrol for 30 min. Western blot showing (**a**) Stat5 (pTyr694), (**b**) Erk (pThr202/Tyr204), and (**c**) Akt (pSer473) expression and total form. Data are shown as the mean ± SEM of three independent experiments. Asterisks indicate statistical significance using one-way ANOVA: ** *p* < 0.01, ns: not significant (*p* > 0.05).

**Figure 3 ijms-21-09575-f003:**
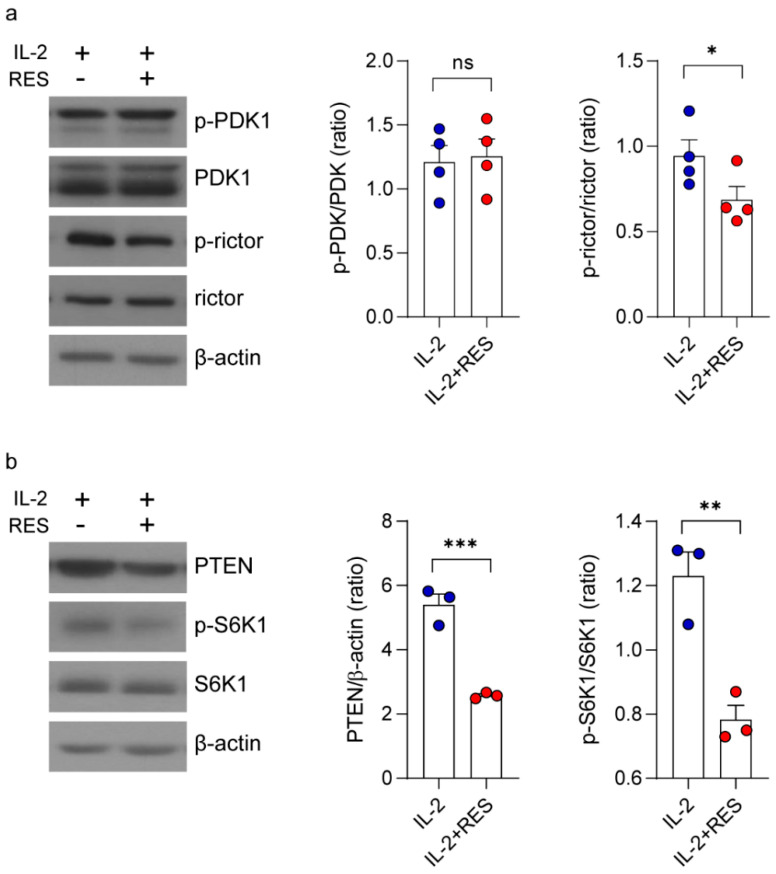
Regulation of the mammalian target of rapamycin (mTOR) complex 2 signaling pathway by resveratrol. NK92 cells were deprived of interleukin (IL)-2 for 24 h and then treated with IL-2 (5 ng/mL) with or without 20 μM resveratrol for 30 min. (**a**) Western blot showing phosphoinositide-dependent kinase-1 (PDK1) (pSer241) and rictor (pThr1135) expression and total form. Data are shown as the mean ± SEM of four independent experiments. (**b**) Western blot showing phosphatase and tensin homolog (PTEN) and ribosomal protein S6 kinase beta-1 (S6K1) (pThr389) expression and total form. Data are shown as the mean ± SEM of three independent experiments. Asterisks indicate statistical significance using the Student’s *t*-test: * *p* < 0.05, ** *p* < 0.01, *** *p* < 0.001, ns: not significant (*p* > 0.05).

**Figure 4 ijms-21-09575-f004:**
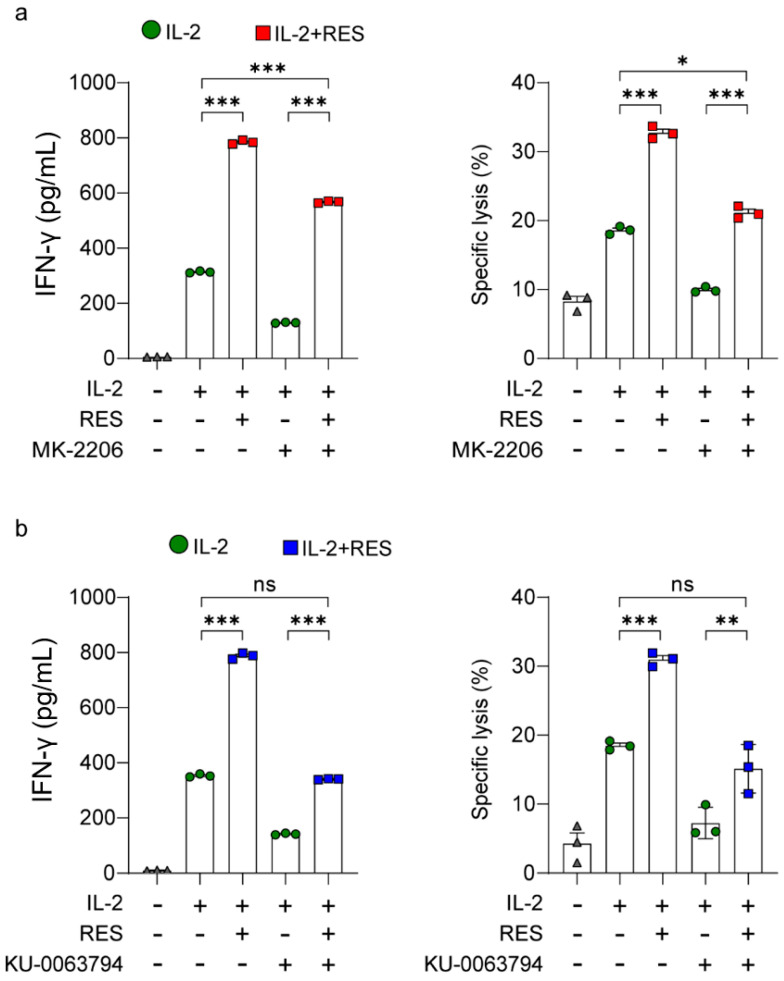
Effects of Akt and mammalian target of rapamycin (mTOR) complex 2 (mTORC2) on natural killer (NK) cell activation by resveratrol. Interferon (IFN)-γ secretion in NK92 cells after the indicated treatments for 48 h determined using an enzyme-linked immunosorbent assay. NK cell cytotoxicity was determined using the calcein-AM assay. NK92 effector cells were harvested after the indicated treatments for 36 h, followed by incubation with K562 target cells at a 1:1 ratio for 4 h. (**a**) An Akt inhibitor (MK-2206) was added 30 min before resveratrol treatment. (**b**) An mTOR inhibitor (KU-0063794) was added 30 min before resveratrol treatment. Results are presented as mean ± SD of triplicate measurements. Three independent experiments were performed. Asterisks indicate statistical significance using one-way ANOVA: * *p* < 0.05, ** *p* < 0.01, *** *p* < 0.001, ns: not significant (*p* > 0.05).

**Figure 5 ijms-21-09575-f005:**
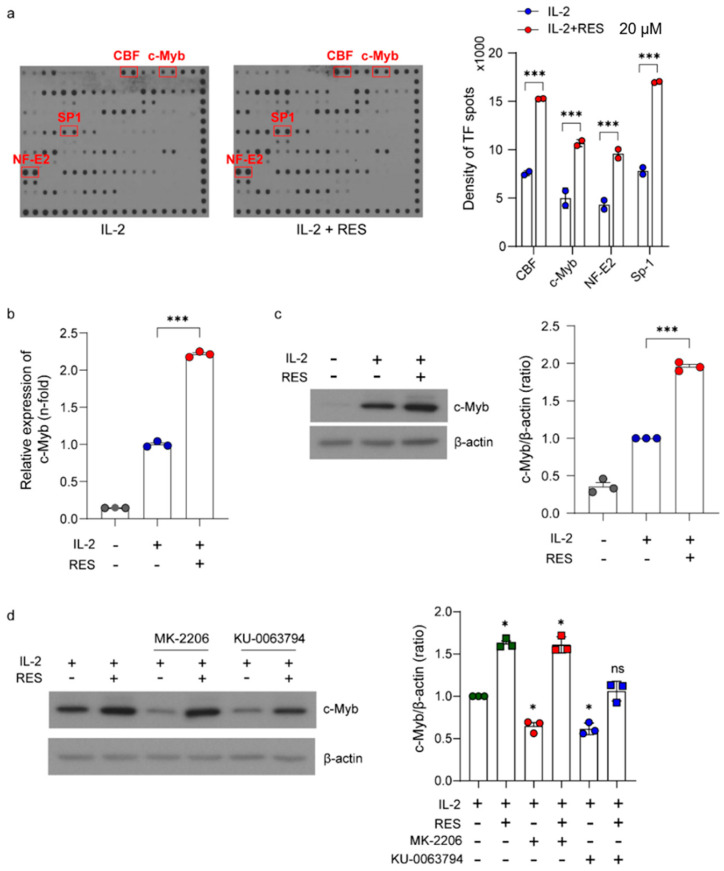
Upregulation of Akt-related transcription factors in natural killer (NK) cells by resveratrol. (**a**) NK92 cells were deprived of interleukin (IL)-2 for 24 h and then treated with IL-2 (5 ng/mL) with or without 20 μM resveratrol for 30 min. Data are shown as the mean ± SD of duplicate measurements. (**b**) The mRNA levels of c-Myb in NK92 cells after resveratrol treatment for 24 h measured using real-time polymerase chain reaction and calculated using the ΔΔCT method. Glyceraldehyde 3-phosphate dehydrogenase (Gapdh) was used as an internal control. (**c**) Western blot showing c-Myb expression. Data are shown as the mean ± SEM of three independent experiments. (**d**) Western blot showing c-Myb expression after treatment with the Akt inhibitor (MK-2206) and mammalian target of rapamycin (mTOR) inhibitor (KU-0063794). Data are shown as the mean ± SEM of three independent experiments. Asterisks indicate statistical significance using one-way ANOVA: * *p* < 0.05, *** *p* < 0.001, ns: not significant (*p* > 0.05).

**Figure 6 ijms-21-09575-f006:**
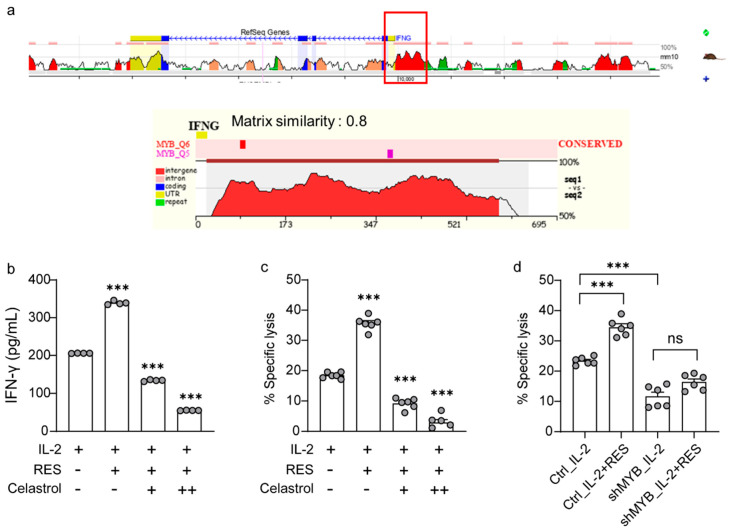
Effects of c-Myb on natural killer (NK) cell activation induced by resveratrol. (**a**) Annotation of conserved transcription factor binding sites underlying the evolutionary conservation of genomes (ECR) browser conservation plots displayed using the “Synteny/Alignments” link in the top menu. (**b**,**c**) The c-Myb inhibitor, celastrol, was added 30 min before resveratrol treatment. (**b**) Interferon (IFN)-γ secretion in NK92 cells after the indicated treatments for 36 h determined using an enzyme-linked immunosorbent assay. Results are presented as mean ± SD of quadruplicate measurements. Three independent experiments were performed. (**c**,**d**) NK cell cytotoxicity was determined using the calcein-AM assay. NK92 effector cells were harvested after the indicated treatments for 36 h, followed by incubation with K562 target cells at a 1:1 ratio for 4 h. Data are shown as the mean ± SEM of six independent experiments. Asterisks indicate statistical significance using one-way ANOVA: *** *p* < 0.001, ns: not significant (*p* > 0.05).

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
