# Peer review of "Resveratrol Activates Natural Killer Cells through Akt- and mTORC2-Mediated c-Myb Upregulation"

_ijms, 2020, doi:10.3390/ijms21249575_

Round 1
Reviewer 1 Report
Very well written article.
Clearly presented research methods and conclusions supported by the results.
Additionally, it can be seen that the article may be used as a starting point for further research, which is even suggested by the authors themselves.
The synergistic effect of resveratrol and IL-2, IL-12 and IL-15 on NK cell activation is clearly visible. It was very appropriate to investigate this relationship. Additionally, regulation of the mammalian target of rapamycin (mTOR) complex 2 signaling pathway by resveratrol was proved It was tested whether NK cell activation is the direct effect of resveratrol-induced Akt and mTORC2 activation, which resulted in proving that NK cell activation via resveratrol is more dependent on the mTOR pathway than on Akt. Moreover, it was proved that resveratrol-induced activation of NK cells is dependent on c-Myb. The above results on NK92 cell lines are very valuable and may give rise to studies on purified NK cells. I hope the authors have such research with intention in the future.
I would ask for the font unification, incl. in e-mails and in degrees Celcius and other places.
Author Response
Thank you for your response and comments on our manuscript.
Responses to Reviewer’s comments (Reviewer #1).
1. I would ask for the font unification, incl. in e-mails and in degrees Celcius and other places.
As the reviewer requested, the fonts were unified in the revised manuscript.
Thank you for your valuable comments.
Sincerely yours,
Jongsun Kim
Reviewer 2 Report
The manuscript by Lee and Kim takes its steps from their previous paper in which resveratrol was identified among antioxidants, vitamins and food ingredients for its ability to activate NK cells. In this manuscript the authors address the mechanisms underlying the activating effect of resveratrol. The authors first demonstrate that resveratrol, while potentiating IFNg production when added concomitantly to IL-2, IL-12 or IL-15, upregulates the cytotoxic activity only in the presence of IL-2. Then, the authors demonstrate that resveratrol activate a signaling pathway that involves Akt and mTOR, the latter playing a major role in the activation of NK cells by resveratrol. The pathway culminates in the activation of c-Myb, that the authors have already shown to be involved in the activation of NK cells in a previous manuscript. In general, the mechanistic dissection of the activating effect of resveratrol is straightforward, the methodological approach is appropriate and the conclusions supported by the results. The major limit is that all the experiments are based on the NK92 cell line. It would have been interesting to see some of the key findings reproduced in purified NK cells, thus overcoming some of the limitations of the cell line.
I have some minor comments that the authors may want to address:
- the experimental setup of Figure 1 is not entirely clear to me. NK92 cells are grown in the presence of 5 ng/ml IL-2, as stated in Materials and Methods, since these cells depend on IL-2 for their growth. Is fresh medium without IL-2 used for the experiments with IL-12 or IL-15? Should this be the case, I have two questions: what is the vitality of NK92 cells after 36h/48h in the absence of IL-2? The original paper on NK92 cells (PMID: 8152260) reports that cells die within 72h in the absence of IL-2. The second question relates to the comparison of the three cytokines (in the absence of resveratrol): is IL-12 as effective as IL-2 in inducing K562 lysis?
- is c-Myb also involved in the potentiation of IFNg production by resveratrol and IL-12 or IL-15? This is especially important for IL-15, that shares two receptor subunits with IL-2, also to explain why resveratrol increase cytolytic activity by IL-2, but not IL-15.
Technical questions:
- I could not find a correspondence between some of blots of the manuscript and the non-published material (for instance, p-PDK1 and PDK1 (figure 3a) or Akt (figure 2c)). Are the unpublished material additional experiments?
- it is not always clear if the figures represent technical replicates or independent experiments. For example, the values in Fig. 4a,b for IFNg, and Fig. 6b are ELISA with 3/4 almost identical values.
Author Response
Thank you for your response and comments on our manuscript.
Responses to Reviewer’s comments (Reviewer #2).
1. The experimental setup of Figure 1 is not entirely clear to me. NK92 cells are grown in the presence of 5 ng/ml IL-2, as stated in Materials and Methods, since these cells depend on IL-2 for their growth. Is fresh medium without IL-2 used for the experiments with IL-12 or IL-15? Should this be the case, I have two questions: what is the vitality of NK92 cells after 36h/48h in the absence of IL-2? The original paper on NK92 cells (PMID: 8152260) reports that cells die within 72h in the absence of IL-2. The second question relates to the comparison of the three cytokines (in the absence of resveratrol): is IL-12 as effective as IL-2 in inducing K562 lysis?
We used fresh medium without IL-2 for the experiments with IL-12 or IL-15. As the reviewer pointed out, we added a sentence to represent treatment condition in the revised manuscript (legend of Fig. 1, line 75).
A1. It has been reported that IL-12 and IL-15 are also important for NK cell activation. As far as we observe, NK92 cells can survive in a medium with either IL-12 or IL-15 even in the absence of IL-2 after 36h/48h.
A2. IL-12 is less effective than IL-2 in inducing K562 lysis. As shown in Fig 1b, IL-12 requires a higher dose than IL-2 and has no synergistic effect with resveratrol.
2. Is c-Myb also involved in the potentiation of IFNg production by resveratrol and IL-12 or IL-15? This is especially important for IL-15, that shares two receptor subunits with IL-2, also to explain why resveratrol increase cytolytic activity by IL-2, but not IL-15.
It was not confirmed in this study whether c-Myb is involved in the potentiation of IFNg production by resveratrol and IL-12 or IL-15. IL-2 and IL-15 have been reported to have different effects, and by guessing from our results, c-Myb is expected to be IL-2 dependent.
Technical questions:
3. I could not find a correspondence between some of blots of the manuscript and the non-published material (for instance, p-PDK1 and PDK1 (figure 3a) or Akt (figure 2c)). Are the unpublished material additional experiments?
No, they are not additional data. In order to ensure the integrity and scientific validity of blots, original, uncropped and unadjusted images were provided as non-published material.
4. It is not always clear if the figures represent technical replicates or independent experiments. For example, the values in Fig. 4a,b for IFNg, and Fig. 6b are ELISA with 3/4 almost identical values.
As the reviewer pointed out, we corrected the figure legends to represent technical replicates in the revised manuscript (legend of Fig. 4a, b and Fig. 6b for IFNg, line 137-138, 191-192).
Thank you for your valuable comments.
Sincerely yours,
Jongsun Kim